# RSO: A Gradient Free Sampling Based Approach For Training Deep Neural Networks

## Abstract

We propose RSO (random search optimization), a gradient free, sampling based approach for training deep neural networks. To this end, RSO adds a perturbation to a weight in a deep neural network and tests if it reduces the loss on a mini-batch. If this reduces the loss, the weight is updated, otherwise the existing weight is retained. Surprisingly, we find that repeating this process a few times for each weight is sufficient to train a deep neural network. The number of weight updates for RSO is an order of magnitude lesser when compared to backpropagation with SGD. RSO can make aggressive weight updates in each step as there is no concept of learning rate. The weight update step for individual layers is also not coupled with the magnitude of the loss. RSO is evaluated on classification tasks on MNIST and CIFAR-10 datasets with deep neural networks of 6 to 10 layers where it achieves an accuracy of 99.1% and 81.8% respectively. We also find that after updating the weights just 5 times, the algorithm obtains a classification accuracy of 98% on MNIST.

## 1 Introduction

Deep neural networks solve a variety of problems using multiple layers to progressively extract higher level features from the raw input. The commonly adopted method to train deep neural networks is backpropagation (Rumelhart et al. (1985)) and it has been around for the past 35 years. Backpropagation assumes that the function is differentiable and leverages the partial derivative w.r.t the weight $w_i$ for minimizing the function $f(x, w)$ as follows,

$$w_{i+1} = w_i - \eta \nabla f(x, w) \Delta w_i$$

, where $\eta$ is the learning rate. Also, the method is efficient as it makes a single functional estimate to update all the weights of the network. As in, the partial derivative for some weight $w_j$, where $j \neq i$ would change once $w_i$ is updated, still this change is not factored into the weight update rule for $w_j$. Moreover, it may not even be optimal for all weights to move in the same direction as obtained from the gradients in the previous layer. Although deep neural networks are non-convex (and the weight update rule measures approximate gradients), this update rule works surprisingly well in practice.

To explain the above observation, recent literature (Du et al. (2019); Li & Liang (2018)) argues that because the network is over-parametrized, the initial set of weights are very close to the final solution and even a little bit of nudging using gradient descent around the initialization point leads to a very good solution. We take this argument to another extreme - instead of using gradient based optimizers - which provide strong direction and magnitude signals for updating the weights; we explore the region around the initialization point by sampling weight changes to minimize the objective function. Formally, our weight update rule is

$$w_{i+1} = \begin{cases} w_i, & f(x, w_i) <= f(x, w_i + \Delta w_i) \\ w_i + \Delta w_i, & f(x, w_i) > f(x, w_i + \Delta w_i) \end{cases}$$

, where $\Delta w_i$ is the weight change hypothesis. Here, we explicitly test the region around the initial set of weights by computing the function and update a weight if it minimizes the loss, see Fig. 1.

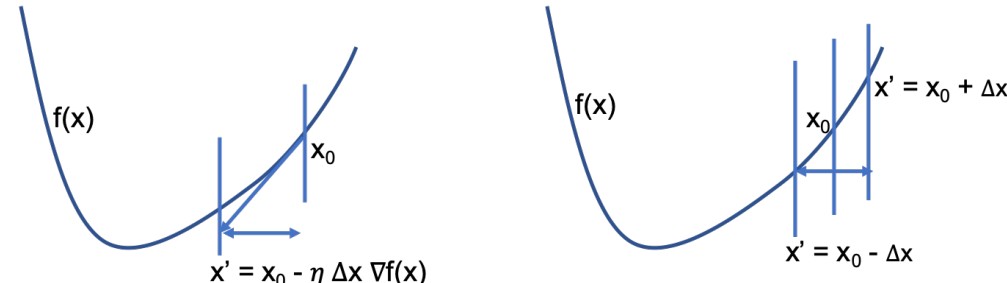

Figure 1: Gradient descent vs sampling. In gradient descent we estimate the gradient at a given point and take a small step in the opposite direction direction of the gradient. In contrast, for sampling based methods we explicitly compute the function at different points and then chose the point where the function is minimum.

Surprisingly, our experiments demonstrate that the above update rule requires fewer weight updates compared to backpropagation to find good minimizers for deep neural networks, strongly suggesting that just exploring regions around randomly initialized networks is sufficient, even without explicit gradient computation. We evaluate this weight update scheme (called RSO; random search optimization) on classification datasets like MNIST and CIFAR-10 with deep convolutional neural networks (6-10 layers) and obtain competitive accuracy numbers. For example, RSO obtains 99.1% accuracy on MNIST and 81.8% accuracy on CIFAR-10 using just the random search optimization algorithm. We do not use any other optimizers for optimizing the final classification layer.

Although RSO is computationally expensive (because it requires updates which are linear in the number of network parameters), our hope is that as we develop better intuition about structural properties of deep neural networks, we will be able to accelerate RSO (using Hebbian principles, Gabor filters, depth-wise convolutions). If the number of trainable parameters are reduced drastically (Frankle et al. (2020)), search based methods could be a viable alternative to back-propagation. Furthermore, since architectural innovations which have happened over the past decade use back-propagation by default, a different optimization algorithm could potentially lead to a different class of architectures, because minimizers of an objective function via different greedy optimizers could potentially be different.

## 2 RELATED WORK

Multiple optimization techniques have been proposed for training deep neural networks. When gradient based methods were believed to get stuck in local minima with random initialization, layer wise training was popular for optimizing deep neural networks (Hinton et al. (2006); Bengio et al. (2007)) using contrastive methods (Hinton (2002)). In a similar spirit, recent work, Greedy InfoMax by Löwe et al. (2019) maximizes mutual information between adjacent layers instead of training a network end to end. Taylor et al. (2016) finds the weights of each layer independently by solving a sequence of optimization problems which can be solved globally in closed form. Weight perturbation (Werfel et al. (2004)) based methods have been used for approximate gradients estimation in situations where gradient estimation is expensive. However, these training methods do not generalize to deep neural networks which have more than 2-3 layers and its not shown that the performance increases as we make the network deeper. Hence, back-propagation with SGD or other gradient based optimizers (Duchi et al. (2011); Sutskever et al. (2013); Kingma & Ba (2014)) are commonly used for optimizing deep neural networks.

Recently, multiple works have proposed that because these networks are heavily over-parametrized, the initial set of random filters is already close to the final solution and gradient based optimizers only nudge the parameters to obtain the final solution (Du et al. (2019); Li & Liang (2018)). For example, only training batch-norm parameters and keeping the random filters fixed can obtain very good results with heavily parametrized very deep neural networks ($> 800$ layers) as shown in Frankle et al. (2020). It was also shown that networks can be trained by just masking out some weights without modifying the original set of weights by Ramanujan et al. (2020) - although one can argue that

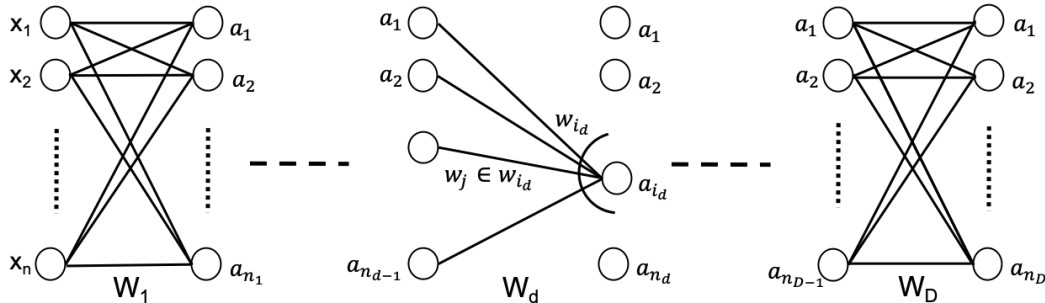

Figure 2: Notation for variables in the network architecture are shown in this figure.

masking is a very powerful operator and can be used to represent an exponential number of output spaces. The network pruning literature covers more on optimizing subsets of an over-parametrized randomly initialized neural network (Frankle & Carbin (2019); Li et al. (2017)). Our method, RSO, is also based on the hypothesis that the initial set of weights is close to the final solution. Here we show that gradient based optimizers may not even be necessary for training deep networks and when starting from randomly initialized weights, even search based algorithms can be a feasible option.

Recently, search based algorithms have gained traction in the deep learning community. Since the design space of network architectures is huge, search based techniques are used to explore the placement of different neural modules to find better design spaces which lead to better accuracy ( Zoph & Le (2016); Liu et al. (2018)). This is done at a block level and each network is still trained with gradient descent. Similar to NAS based methods, weight agnostic neural networks (WANN) (Gaier & Ha (2019)) also searches for architectures, but uses a fixed set of weight values $[-2, -1, -0.5, +0.5, +1, +2]$. WANN operates at a much finer granularity as compared to NAS based methods while searching for connections between neurons and does not use gradient descent for optimization. Algorithms like Deep Neuroevolution by Such et al. (2017) and evolution strategies (ES) by Salimans et al. (2017) are search based optimization algorithms which have been used for training neural networks for reinforcement learning. ES is comprehensively reviewed by Beyer & Schwefel (2004). Both Deep Neuroevolution and Salimans et al. (2017) create multiple replicas (children) of an initial neural network by adding small perturbations to *all* the weight parameters and then update the parameters by either selecting the best candidate or by performing a weighted average based on the reward. Both the methods update all the parameters of the network in each update. The problem with changing all the weights is that updating all the parameters of the network at once leads to random directions which are unlikely to contain a direction which will minimize the objective function and slows down learning (results shown in Section 4.5). Also, these methods were only trained on networks with 2-3 hidden layers, which is fairly shallow when compared to modern deep architectures.

## 3 APPROACH

Consider a deep neural network with $D$ layers, where the weights of a layer $d$ with $n_d$ neurons is represented by $W_d = \{w_1, w_2.., w_{i_d}, ..w_{n_d}\}$. For an input activation $A_{d-1} = \{a_1, a_2, ...a_{n_{d-1}}\}, W_d$ generates an activation $A_d = \{a_1, a_2, ...a_{n_d}\}$, see Fig 2. Each weight tensor $w_{i_d} \in W_d$ generates an activation $a_i \in A_d$, where $a_i$ can be a scalar or a tensor depending on whether the layer is fully connected, convolutional, recurrent, batch-norm etc. The objective of the training process is to find the best set of weights $W$, which minimize a loss function $\mathcal{F}(\mathcal{X}, \mathcal{L}; W)$ given some input data $\mathcal{X}$ and labels $\mathcal{L}$.

To this end, we initialize the weights of the network with a Gaussian distribution $\mathcal{N}(0, \sqrt{2/|w_{i_d}|})$, like He et al. (2015). The input data is also normalized to have zero mean and unit standard deviation. Once the weights of all layers are initialized, we compute the standard deviation $\sigma_d$ of all elements in the weight tensor $W_d$. In the weight update step for a weight $w_j \in w_{i_d}$, $\Delta w_j$ is sampled from $\sim \mathcal{N}(0, \sigma_d)$. We call this $\Delta W_j$ which is zero for all weights of the network but for $w_j$, where

---

**Algorithm 1:** Random Search Optimization

| | |
|---|---|
| **Input** : $\mathcal{X}, \mathcal{L}, C, W = \{W_1, W_2...W_D\}$, | **8 for** $c = 1$ *to* $C$ **do** |
| where $W_d = \{w_1, w_2.., w_{i_d}, ..w_{n_d}\}$ | **9** $\quad$ $d \leftarrow D$ |
| **Output :** $W$ | **10** $\quad$ **while** $d > 0$ **do** |
| **1** $\mathcal{X} \leftarrow \frac{\mathcal{X} - \Sigma X}{\sigma(\mathcal{X})}$ | **11** $\quad\quad$ **for** $w_j \in w_{i_d}$, *where* $w_{i_d} \in W_d$ **do** |
| **2 for** $W_d \in W$ **do** | **12** $\quad\quad\quad$ $x, l \sim \mathcal{X}, \mathcal{L}; \Delta w_j \leftarrow \mathcal{N}(0, \sigma_d);$ |
| **3** $\quad$ **for** $w_j \in w_{i_d}$, *where* $w_{i_d} \in W_d$ **do** | **13** $\quad\quad\quad$ $W \leftarrow argmin(\mathcal{F}(x, l, W + \Delta W_j),$ |
| **4** $\quad\quad$ $w_j \leftarrow \mathcal{N}(0, \sqrt{2/|w_{i_d}|})$ | **14** $\quad\quad\quad\quad$ $\mathcal{F}(x, l, W), \mathcal{F}(x, l, W - \Delta W_j))$ |
| **5** $\quad$ **end** | **15** $\quad\quad$ **end** |
| **6** $\quad$ $\sigma_d \leftarrow \sigma(W_d)$ | **16** $\quad\quad$ $d \leftarrow d - 1$ |
| **7 end** | **17** $\quad$ **end** |
| | **18 end** |
| | **19 return** $W$ |

---

$j \in i_d$. For a randomly sampled mini-batch $(x, l) \in \mathcal{X}, \mathcal{L}$, we compute the loss $\mathcal{F}(x, l; W)$ for $W$, $W + \Delta W_j$ and $W - \Delta W_j$. If adding or subtracting $\Delta W_j$ reduces $\mathcal{F}$, $W$ is updated, otherwise the original weight is retained. This process is repeated for *all* the weights in the network, i.e., to update all the weights of the network once, $\mathcal{F}$ needs to be computed three times the number of weight parameters in the network, $3|W|$. We first update the weights of the layer closest to the labels and then sequentially move closer to the input. This is typically faster than optimizing the other way, but both methods lead to good results. This algorithm is described in Algorithm 1.

In Algorithm 1, in line 12, we sample change in weights from a Gaussian distribution whose standard deviation is the same as the standard deviation of the layer. This is to ensure that the change in weights is within a small range. The Gaussian sampling can also be replaced with other distributions like uniform sampling from $(-2\sigma_d, 2\sigma_d)$ or just sampling values from a template like $(-\sigma_d, 0, \sigma_d)$ and these would also be effective in practice. The opposite direction of a randomly sampled weight is also tested because often it leads to a better hypothesis when one direction does not decrease the loss. However, in quite a few cases (close to 10% as per our experiments), not changing the weight at all is better. Note that there is no concept of learning rate in this algorithm. We also do not normalize the loss if the batch size increases or decreases as the weight update step is independent of the magnitude of the loss.

There is widespread belief in the literature that randomly initialized deep neural networks are already close to the final solution (Du et al. (2019); Li & Liang (2018)). Hence, we use this prior and explore regions using bounded step sizes ($\mathcal{N}(0, \sigma_d)$) in a single dimension. We chose to update one weight at a time instead of sampling all the weights of the network as this would require sampling an exponential number of samples to estimate their joint distribution. RSO will be significantly faster even if prior knowledge about the distribution of the weights of individual neurons is used.

## 4 EXPERIMENTS

We demonstrate the effectiveness of RSO on image classification tasks by reporting the accuracy on the MNIST (Lecun et al. (1998)) and the CIFAR-10 (Krizhevsky (2009)) data sets. MNIST consists of 60k training images and 10k testing images of handwritten single digits and CIFAR-10 consists of 50k training images and 10k testing images for 10 different classes of images.

### 4.1 MNIST

We use a standard convolution neural network (CNN) with 6 convolution layers followed by one fully connected layer as the baseline network for MNIST. All convolution layers use a $3 \times 3$ filter and generate an output with 16 filters channels. Each convolution layer is followed by a Batch Norm layer and then a ReLU operator. Every second convolution layer is followed by a $2 \times 2$ average pool operator. The final convolution layer's output is pooled globally and the pooled features are input to a fully connected layer that maps it to the 10 target output classes. The feature output is mapped to probability values using a softmax layer and cross entropy loss is used as the objective function when choosing between network states. For RSO, we train the networks for 50 cycles as described

| | Random Search (ours) | Backpropagation (SGD) | WANN Gaier & Ha (2019) |
|---|---|---|---|
| Accuracy | 99.12 | 99.27 | 94.2 |

Table 1: Accuracy of RSO on the MNIST data set compared with back propagation and WANN. The CNN architecture used for RSO and SGD is described in 4.1.

in Section 3 and in each cycle the weights in each layer are optimized once. We sample random 5000 samples in a batch per weight for computing the loss during training. The order of updates within each layer is discussed in Section 4.3. After optimizing a convolution layer $d$, we update the parameters of it's batch norm layer using search based optimization as well.

For layer $d$ at cycle $c$, we sample weight updates from a Gaussian with mean 0 and standard deviation $\sigma_d^c$. At the first cycle, this standard deviation is set to the standard deviation of the layer at initialization $\sigma_d$. We linearly anneal the standard deviation of the Gaussian such that at the final cycle $C$, $\sigma_d^C = \sigma_d^1/10$. RSO is robust to the choice of the initial value of the standard deviation. On varying the initial value from $0.1\sigma_d$ to $10\sigma_d$, the minimum accuracy on MNIST is within $0.15$ of the maximum accuracy. In Table 1 we compare the performance of RSO with training using backpropagation (with SGD) and with the training based approach described in Gaier & Ha (2019). The network is able to achieve a near state-of-the-art accuracy of 99.12% using random search alone to optimize all the weights in the network.

## 4.2 SAMPLING MULTIPLE WEIGHTS VERSUS A SINGLE WEIGHT

We compare different strategies each with different set of weights that are perturbed at each update step to empirically demonstrate the effectiveness of updating the weights one at a time (Algorithm 1). The default strategy is to sample a single weight per update step and cycle through the layers and through each weight within each layer. A second possible strategy is to optimize *all* the weights in a *layer* jointly at each update step and then cycle through the layers. The third strategy is to sample *all* the weights in whole *network* at every update step. For layer-level and network-level sampling, we obtain optimal results when the weight changes for a layer $d$ with weight tensor $W_d$ are sampled from $\sim \mathcal{N}(0, \sigma_d/10)$, where $\sigma_d$ is the standard deviation of all elements in $W_d$ after $W_d$ is initialized for the first time. We optimize each of the networks for 500K steps for each of the three strategies and report performance on MNIST in Figure 2. For the baseline network described in Section 4.1, 500K steps translate to about $42$ cycles when using the single weight update strategy. Updating a single weight obtains much better and faster results compared to layer level random sampling which, in turn, is faster compared to a network-level random sampling. When we test on harder data sets like CIFAR-10, the network-level and layer-level strategies do not even show promising initial accuracies when compared to the single weight sampling strategy. The network level sampling strategy is close to how genetic algorithms function, however, changing the entire network is significantly slower and less accurate that RSO. Note that our experiments are done with neural networks with 6-10 layers.

## 4.3 ORDER OF OPTIMIZATION WITHIN A LAYER

When individually optimizing all the weights in a layer $W_d$, RSO needs an order for optimizing each of the weights, $w_j \in w_{i_d}$, where $n_d$ is the number of neurons and $W_d = \{w_1, w_2.., w_{i_d}, ..w_{n_d}\}$. $n_d$ is the number of output channels in a convolution layer and each neuron has $L \times k \times k$ weights, where $L$ the number of input channels and $k$ is the filter size. By default, we first optimize the set of weights that affect each output neuron $w_{i_d} \in W_d$ and optimize the next set and so on till the last neuron. Similarly, in a fully connected layer we first optimize weights for one output neuron $w_{i_d} \in W_d$ and then move to the next in a fixed manner. These orders do not change across optimization cycles. This ordering strategy obtains 99.06% accuracy on MNIST. To verify the robustness to the optimization order, we inverted the optimization order for both convolution and fully connected layers. In the inverted order, we first optimize the set of weights that interact with one input channel $L_i \in L$ and then move to the next set of weights and so on. Inverting the order of optimization leads to a performance of 99.12% on the MNIST data set. The final performance for the two runs is almost identical and demonstrates the robustness of the optimization algorithm to a given optimization order in a layer.

|         | 100K Steps | 200K Steps | 300K Steps | 400K Steps | 500K Steps | 600K Steps |
|---------|------------|------------|------------|------------|------------|------------|
| Network | 88.7       | 91.81      | 91.89      | 92.71      | 93.01      | 94.49      |
| Layer   | 94.85      | 95.77      | 96.95      | 97.24      | 97.25      | 97.53      |
| Weight  | 98.24      | 98.78      | 98.94      | 99.0       | 99.08      | 99.12      |

Table 2: Accuracy of RSO on the MNIST data set reported at different stages of training while sampling at different levels - at the network level, at the layer level and at the weight level. The final performance and the rate of convergence when sampling at the weight level is significantly better than other strategies.

| Stage    | conv1             | conv2                        | conv3                        | conv4                               | Accuracy |
|----------|-------------------|------------------------------|------------------------------|-------------------------------------|----------|
| Depth-3  | -                 | -                            | $3 \times 3, 32$             | $3 \times 3, 32$                    | 56.30%   |
| Depth-5  | $3 \times 3, 16$  | $3 \times 3, 16$             | $3 \times 3, 32$             | $3 \times 3, 32$                    | 68.48%   |
| Depth-10 | $3 \times 3, 16$  | $\langle 3 \times 3, 16 \rangle$ | $\langle 3 \times 3, 32 \rangle$ | $\langle 3 \times 3, 32 \rangle \times 2$ | 81.80%   |

Table 3: Accuracy of RSO with architectures of varying depth on CIFAR-10. The $\langle \cdot \rangle$ brackets represent a basic residual block (He et al. (2016)) which contains two convolution layers per block and a skip connection across the block. Each convolution layer is represented as the filter size and the number of output filters. The conv2 and conv3 stages are followed by a $2 \times 2$ average pooling layer. The accuracy increases as the depth increases which demonstrates the ability of random search optimization to learn well with deeper architectures.

## 4.4   CIFAR-10

On CIFAR-10, we show the ability of RSO to leverage the capacity of reasonably deep convolution networks and show that performance improves with an increase in network depth. We present results on architectures with 2, 4 and 9 convolution layers followed by one fully connected layer. The CNN architectures are denoted by Depth-$l$, where $l$ is the number of convolution layers plus 1 and the details of the architectures are reported in Table 3. Each convolution layer is followed by a Batch Norm layer and a ReLU activation. The final convolution layer output is pooled globally and the pooled features are input to a fully connected layer that maps it to the 10 target output classes. For RSO, we train the networks for 500 cycles and in each cycle the weights in each layer are optimized once as described in Section 3. To optimize each weight we use a random batch of 5000 samples. The performance of RSO improves significantly with increase in depth. This clearly demonstrates that RSO is able to leverage the improvements which come by increasing depth and is not restricted to working with only shallow networks. A comparison for different depth counts is shown in Table 3.

To compare with SGD, we find the hyper-parameters for the best performance of the Depth-10 architecture by running grid search over batch size from 100 to 20K and learning rate from 0.01 to 5. In RSO, we anneal the standard deviation of the sampling distribution, use weight decay and do not use any data augmentation. For SGD we use a weight decay of 0.0001, momentum at 0.9, no data augmentation and step down the learning rate by a factor of 10 twice. The top performance on CIFAR-10 using the Depth-10 network was 82.94%.

## 4.5   COMPARISON OF TOTAL WEIGHT UPDATES

RSO samples each weight once per cycle and a weight may or may not be updated. For $C$ cycles, the maximum number of times all weights are updated is $C$. Back-propagation based SGD updates each weight in the network at each iteration. For a learning schedule with $E$ epochs and $B$ batches per epoch, the total number of iterations is $E \times B$. On the left in Figure 3 we report the accuracy versus the number of times all the weights are updated for RSO and SGD on the MNIST data set. For SGD, we ran grid search as described in Section 4.4 to find hyper-parameters that require the minimum steps to reach $\geq 99.12\%$ accuracy because that is the accuracy of RSO in 50 cycles. We found that using a batch size of 5000, a learning rate of 0.5 and a linear warm-up for the first 5 epochs achieves 99.12% in less than 600 steps.

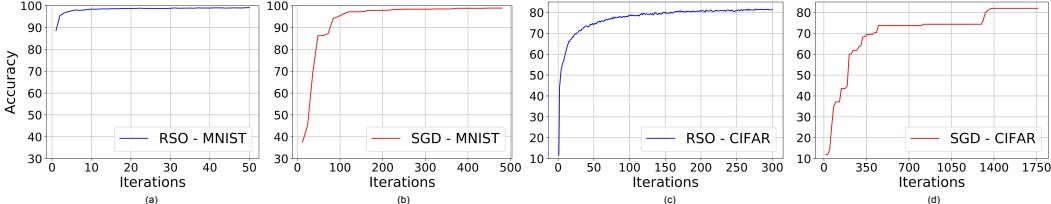

Figure 3: Accuracy versus the number of times all the weights are updated for RSO and SGD on the MNIST and on the CIFAR-10 data set. The results demonstrate that the number of times all the weights need to be updated in RSO is typically much smaller than the corresponding number in SGD. Note that since the weight update step for RSO is linear in the number of parameters of the network, each weight update step is significantly more expensive. For example, on MNIST, SGD takes 10 seconds per epoch while RSO takes 12 minutes. On CIFAR10, RSO takes 52 minutes per epoch while SGD takes 29 seconds.

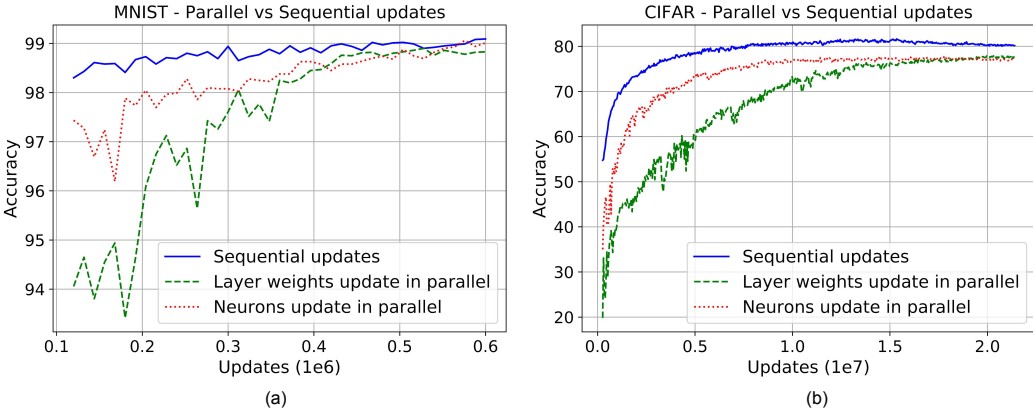

Figure 4: Sequential updates versus updating all the weights in convolution layers in parallel or updating the neurons in parallel. Sequential updates perform the best for both data sets. The parallel update scheme enables using distributed learning techniques to reduce wall-clock time to reach convergence for MNIST. For CIFAR-10, the gap between sequential and parallel may be closed by running the parallel setup on a longer learning schedule using distributed learning hardware.

On the right in Figure 3 we report the accuracy on the Depth-10 network (Table 3) at different stages of learning on CIFAR-10 for RSO and SGD. We apply grid search to find optimal hyper-parameters for SGD such that the result is $\geq 81.8\%$, which is the accuracy of RSO on the Depth-10 network after 300 cycles. The optimal settings use a batch size of 3000, a learning rate of 4.0, momentum at 0.9 and a linear warm-up for the first 5 epochs to achieve an accuracy of $81.98\%$ in a total of 1700 iterations. The results on number of iterations demonstrate that the number of times all the weights are updated in RSO, $C$, is typically much smaller than the corresponding number in SGD, $E \times B$ for both MNIST and CIFAR-10. Further, RSO reaches $98.0\%$ on MNIST after updating all the weights *just* 5 times, which indicates that the initial set of random weights is already close to the final solution and needs small changes to reach a high accuracy.

## 4.6 Updating weights in parallel

RSO is computationally expensive and sequential in its default update strategy, section 4.3. The sequential strategy ensures that the up to date state of the rest of the network is used when sampling a single weight in each update step. In contrast, the update for each weight during an iteration in the commonly used backpropagation algorithm is calculated using gradients that are based on a single state of the network. In backpropagation, the use of an older state of the network empirically proves to be a good approximation to find the weight update for multiple weights. As shown in section 4.2, updating a single weight at each step leads to better convergence with RSO as compared to updating

weights jointly at the layer and network level. If we can search for updates by optimizing one weight at a time and use an older state of the network, we would have an embarrassingly parallel optimization strategy that can then leverage distributed computing to significantly reduce the run time of the algorithm.

We found that the biggest computational bottleneck was presented by convolution layers and accordingly experimented with two levels of parallel search specifically for convolution layers. The first approach was to search each of the weights in a layer $w_j \in w_{i_d}, W_d = \{w_1, w_2.., w_{i_d}, ..w_{n_d}\}$ in parallel. The search for the new estimate for each weight uses the state of the network before the optimization started for layer $d$. The second approach we tried was to search for weights of each output neuron $w_{i_d}$ in parallel. We spawn different threads per neuron and within each thread we update the weights for the assigned neuron sequentially. Finally we merge the weights for all neurons in the layer. Results on the MNIST data set are shown on the left and results on the CIFAR-10 data set are shown on the right in Figure 4. Regarding the rate of convergence, sequential updates outperform both layer-level parallel and updating the neurons in parallel. However, both parallel approaches converge almost as well as sequential search at the end of 50 cycles on MNIST. For the CIFAR-10 data set, the sequential update strategy seems to out perform the parallel search strategies by a significant margin over a learning schedule of 500 cycles. However, given the embarrassingly parallel nature of both parallel search strategies, this limitation may be overcome by running the experiment on a longer learning schedule using distributed learning hardware to possibly close this gap. This is a hypothesis which will have to be tested experimentally in a distributed setting.

### 4.7 CACHING NETWORK ACTIVATION

Throughout this paper, we have described neural networks with $D$ convolution layers and 1 fully connected layer. The number of weight parameters $|W| = n_D \times 10 + \sum_1^D n_d \times i_d \times k_d \times k_d$, where $n_d, i_d$ and $k_d$ are the number of output filters, number of input filters and filter size respectively, for convolution layer $d \in D$. The FLOPs of a forward pass for a batch size of 1 is $F = n_D \times 10 + \sum_1^D n_d \times i_d \times k_d \times k_d \times s_d \times s_d$, where $s_d$ is the input activation size at layer $d$. Updating all the parameters in the network once using a batch size 1 requires $3 \times |W| \times F$ FLOPs, which leads to a computationally demanding algorithm.

To reduce the computational demands for optimizing a layer $d$, we cache the network activations from the layer $d-1$ before optimizing layer $d$. This enables us to start the forward pass from the layer $d$ by sampling a random batch from these activations. The FLOPs for a forward pass starting at a convolution layer $d$ reduce to $F_d = n_D \times 10 + \sum_d^D n_d \times i_d \times k_d \times k_d \times s_d \times s_d$. The cost of caching the activations of layer $d-1$ for the complete training set is negligible if computed at a batch size large enough that the number of forward iterations $I_d << |W_d| = n_d \times i_d \times k_d \times k_d$. The FLOPs for training the parameters once using this caching techniques is $\sum_1^D 3 \times |W_d| \times F_d$. This caching strategy leads to an amortized reduction in FLOPs by 3.0 for the MNIST network described in 4.1 and by 3.5 for the Depth-10 network in Table 3.

## 5 CONCLUSION AND FUTURE WORK

RSO is our first effective attempt to train reasonably deep neural networks ($\geq 10$ layers) with search based techniques. A fairly naive sampling technique was proposed in this paper for training the neural network. Yet, as can be seen from the experiments, RSO converges an order of magnitude faster compared to back-propagation when we compare the number of weight updates. However, RSO computes the function for each weight update, so its training time scales linearly with the number of parameters in the neural network. If we can reduce the number of parameters, RSO will train faster.

Another direction for speeding up training would be to use effective priors on the joint distribution of weights and sampling using techniques like Blocked Gibbs sampling. While sampling at the layer and network level lead to a drop in performance for RSO, a future direction is to identify highly-correlated and coupled blocks of weights in the network (like Hebbian priors) that can be sampled jointly to reduce computational costs similar to blocked Gibbs. The north star would be a neural network which consists of a *fixed* set of basis functions that need a small set of parameters to

modulate the responses. If we can construct and train such networks with search based methods, it would significantly improve our understanding of deep networks.

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
