# OpenReview forum: "RSO: A Gradient Free Sampling Based Approach For Training Deep Neural Networks"
_ICLR.cc/2021/Conference — Reject_

### Official Review · AnonReviewer4 · 2020-10-29
**I enjoyed this paper, and believe it is the seed of something very important. The main result of a viable alternative to backprop may very well be one of the things that pushes AI into the next stage.  My detailed comments are meant to strengthen the paper.  However, the authors answered nearly all of my questions I had while reading.**

**Rating:** 8
**Confidence:** 5

**Review:**

This paper discusses a possible method for training a deep neural network without using backpropagation.  Backpropagation has been very successful in minimizing output errors for both supervised and unsupervised methods and has stood up to challenges from other methods.  The motivation for finding suitable replacements or approximations is to reduce the computational complexity of training neural networks.  This can take the form of reducing the size of the NN required to accomplish a task or simply train an NN with a smaller number of operations. I believe this is a very important new topic to find viable alternatives to backprop.  These kinds of methods have advantages on better-utilizing memory bandwidth, making cheaper hardware more relevant to the training side of NNs.

The authors do a good job of giving background by citing node perturbation methods, lottery ticket hypothesis, and genetic methods.  They all appear to be pointing to an underlying concept that random initializations in overparameterized networks already have embedded, sparse representations.

The main result of the paper is that a small number of sequential weight updates using the authors' proposed algorithm rivals the performance of an established method like backpropagation.  The proposed algorithm is simply to perturb weights from a randomly initialized neural network and keep the perturbation if it reduces the loss on a minibatch. This relies on an assumption that a randomly initialized network is close to a final solution.

I really enjoyed this paper. Nearly every question I asked myself while reading it was answered in a subsequent section.  As pointed out, this is the first step at a new concept.  As with any good paper, this paper begets a lot more questions than it completely answers.

Suggestions:
Section 3: what is the motivation for using a Gaussian distribution to initialize weights?  Not that I see anything wrong with that, but is there some reason this might be better or worse than other initializations?
Section 3: “We first update the weights of the layer closest…”.  This could be an area of additional research as to where to update first.  If we look at neuroscience, we see that layers closer to inputs seems to learn first, so might be good to include some discussion on that here.
Section 4: These are good networks to start with, but I would like to see larger networks that are more relevant to problems today….transformers being trained to useful levels using this method could be a huge and important step.

Section 4.1: It could strengthen the paper to include some analysis on the number of MAC operations required and the number of reads/writes to memory for SGD vs RSO.  This could be useful in this paper, or a subsequent one.

Section 4.2: Some theory likely needs to be developed here.  It would good to add some discussion about the tradeoffs between these options. I believe this is more for future work.

Section 4.5: If the RSO algorithm is more amenable to parallelism, that could be an important advantage.  Some discussion of that vs SGD could also build a stronger case.ZS

---

> ### Author Response · Authors · 2020-11-18
> **Reply to AnonReviewer4**
>
> Thank you for appreciating our research with a highly positive review.
>
> Section 3: Weight initialization is performed with a random normal distribution as mentioned in Delving deep into rectifiers: Surpassing human level performance on imagenet classification, by He. et. al 2015. It provides a theoretically sound way of initializing deep neural networks which have ReLU non-linearities. We tried to train networks from the input layer, however, we found that starting from the output layers was slightly better in the initial stage of the learning process. This is because final layers can influence the loss by a larger margin at the very beginning of training. We agree that order of updates needs a closer look and ideas from neuroscience need to be incorporated into learning algorithms.
>
> Section 4. Computational requirements and accelerating RSO:
> We did not consider computational budgets while designing RSO as we were not even sure if such an algorithm would ever converge on image recognition datasets. In its current state, although RSO may require 100 times less iterations than SGD, one has to keep in mind that even a small neural network contains hundreds of thousands of parameters. State-of-the-art neural networks (like transformers) can contain millions of parameters. So, in its current form, the number of MAC operations needed for RSO would be 100 to 1000x times larger compared to SGD, depending on the type of architecture. However, as we discover more about properties of neural networks and design architectures with extremely less parameters (like ShuffleNet or https://openreview.net/forum?id=vYeQQ29Tbvx where we only learn batch-norm parameters), RSO could be a viable alternative for methods like SGD. Training only BN parameters can accelerate training with RSO by 2-3 orders of magnitude. Because of different optimization techniques, we also hope that newer architectures may be discovered which obtain better performance with lesser computational budgets.
>
> RSO also has a benefit over SGD in terms of memory consumption as it only performs forward propagation, so memory consumption is bounded by the maximum amount of memory a layer takes during training. Since RSO only needs to perform forward propagation, we can use lower precision activations like 4 or 8 bit during inference which can boost the performance by 5-10 times (and also reduce memory), depending on the level of quantization used. Because each weight update can be made in parallel, with the parallel weight update scheme, RSO is extremely easy to parallelize on different GPUs and hosts when compared to SGD. We have a discussion on this topic in Section 4.6. However, because we did not perform the experiments in a distributed setting to validate the accuracy, we did not make strong claims about parallelization in the paper.
>
> Section 4.2. While sampling at a neuron level, one can use priors from the data distribution which can lead to faster convergence. Currently, RSO does not use any prior information about the distribution of activations at a given layer. We have only discussed a very naive strategy for updating the weights in parallel. This is definitely an area of research for future work and we are pursuing these directions.

---

> > ### Comment · AnonReviewer4 · 2020-11-21
> > **Precision of statements**
> >
> > Thanks for incorporating the feedback.  While I don't agree with the rejection above, the reviewer's comments should be taken seriously.  Precision in statements is important and will make a stronger paper.  Please consider those updates.  The central idea of the paper is good and making it very clear what the novel innovations are should be your goal.

---

> > > ### Author Response · Authors · 2020-11-21
> > > **Precision of statements, response**
> > >
> > > We had updated our paper taking the feedback into account, for example citations for ES etc. The related work section was echoing some of the recent literature which discussed that gradient descent leads to global minima for certain type of deep networks. As the statement is controversial, we have removed that line from related work. Some other statements in the related work section have also been updated. We still feel the statements in the related work were not incorrect, as we backed them with citations, but they were debatable (which we clearly don't want), so have made the changes. Overall, we agree that these changes will improve the paper but would strongly recommend that acceptance or rejection of this paper not be made on these details, as these are minor changes, which can be corrected.

---

### Official Review · AnonReviewer1 · 2020-10-29
**Very simple but potentially very practical idea.**

**Rating:** 7
**Confidence:** 4

**Review:**

In this paper, rather than training a DNN using SGD, the proposed idea is to perturb the weights of the network and accept the perturbation if it improves the performance. This naive idea seems to perform almost as well as SGD, and "an order of magnitude faster" (see below).

While I commend the authors for bringing up the fact that such training is possible (which can be very practical in RAM bound settings). There are at least two things that I would like to see fixed:

- Markov Chain Monte Carlo is mentioned in the abstract and never discussed again. Though the method certainly resembles MCMC to some degree, either you make this connection explicit and say under which probabilistic model this approach corresponds to MCMC, and potentially connect the values sampled over time with a posterior density, or eliminate the references to MCMC. As of now, you simply mention it but it begs a question more than answers anything.

- Why would it be meaningful to compare SGD and RSO in terms of "cycles", given that cycles have vastly different computational costs for SGD and RSO? In fact, a comparison in terms of cycles would be useful, but using the same update schedule for SGD that you use for RSO (which of course would make SGD, the competing approach, take a higher computational cost per cycle). Right now you are stating that RSO is an order of magnitude faster... when measured in a unit (cycles) that is much more costly for RSO. That statement is meaningless. So a) show a fair comparison, for instance accuracy vs. compute time, using a state-of-the-art, GPU-optimized version of SGD (and also of RSO). b)  Given that a) is done, it would also be useful to show the current version of  accuracy vs. cycle time if you consider the two potential versions of SGD cycles: parallel updates and sequential updates. That might make RSO not seem as good compared with SGD but would be much more useful to judge when RSO is to be preferred to SGD.

Edit: score modified after reading the authors' reply.

Edit 2: Regarding the issue of optimization getting stuck in local optima or finding global minima (moved my comment here for visibility):

I thank the authors for their flexibility on this issue, but I'd like to weigh in on this saying that from my perspective those statements were actually accurate and useful, and should be kept in the paper as they existed originally. They are properly backed up by citations, unlike the comment of "getting stuck for sure in a local optimum", which is very much dependent on the optimization problem. I am willing to follow up with the other reviewers, should they consider I am mistaken.

I have also increased my score after reading the author's response, and I agree that the final decision should not depend on this specific issue. If the paper is accepted, I'd like it to include the original statement, which can be very informative for some readers.

---

> ### Author Response · Authors · 2020-11-18
> **Reply to AnonReviewer1**
>
> Thank you for reviewing our paper. We provide a response to the concerns.
>
> We agree that we did not build upon the Markov Chain Monte Carlo part of the algorithm later in the paper. The only similarity is that we make random moves which are conditioned on previous updates. As mentioned in your review, and going through the paper again, we agree that mentioning MCMC in the abstract does not do justice to the term. We have updated the abstract and do not mention MCMC any more.
>
> Why would it be more meaningful to compare in terms of "cycles"?
> This is done to highlight that it is possible to train deep neural networks in very few weight updates (which can be as less as 5). We think this was an interesting observation when training deep neural nets with RSO and this type of presentation brings to light the importance of explicitly searching for the direction for each weight parameter.
> Computational speed was not a consideration while designing this algorithm and we do not claim that RSO is faster in terms of compute time w.r.t SGD or that RSO should be used instead of SGD for tasks like image classification. We also mention that the computational time for a weight update is linear in the number of weight parameters in the network, so weight update steps are indeed very expensive for RSO. However, as current libraries are heavily optimized with almost a decade of hardware optimization, while RSO is essentially unoptimized python code, we do not have a fair way to compare RSO with SGD, so reporting wall clock times would make the algorithm look even worse than it is. If exact wall clock times are needed, we can tell that it takes 12 minutes per epoch to train with RSO on MNIST while it takes 10 seconds per epoch for SGD. On CIFAR10, SGD takes 29 seconds per epoch while RSO takes 52 minutes. So, even if SGD needs 20x more epochs to converge, it would still be faster. We have added these details below Figure 3. With that said, there are several avenues to pursue research on accelerating RSO as mentioned in response to AnonReviewer4 in the “Computational requirements and accelerating RSO” comment.

---

### Official Review · AnonReviewer2 · 2020-10-29
**Nice work on randomized learning with unclear contribution**

**Rating:** 3
**Confidence:** 5

**Review:**

##########################################################################

Summary:

Instead of back-propagation, the authors consider a randomized search heuristic to train the parameters of neural networks. The proposed work is based on the hypothesis that the initial set of neural network weights is close to the final solution. The authors identify the problem that existing randomized search methods update all the parameters of the network in each update. Thus the proposed method updates only a single weight per iteration. Experimental results on MNIST and CIFAR10 show that the proposed method delivers competitive results.

##########################################################################

Reasons for score:


Overall, I vote for rejecting. Indeed, investigating alternative learning methods for deep architectures is highly relevant. My major concern is about the novelty of the paper and formal presentation (see cons below). I do not expect that the authors can address my concern in the rebuttal period.


##########################################################################Pros:

STRONG

1. Investigating alternative learning methods for deep architectures is highly relevant.

2. Nice practical implementation details are provided like parallel computation or clever caching.

3. This paper provides experiments on well-known benchmark data sets. The results suggest that randomized search heuristics can work well for training the weights of deep neural networks.


##########################################################################

WEAK

1. Highly relevant theoretical work in this field is not referenced or discussed, e.g., Nesterov's "Efficiency of coordinate descent methods on huge-scale optimization problems" or work on randomized search in general, like "Evolution strategies - A comprehensive introduction" by Hans-Georg Beyer and Hans-Paul Schwefel.

2. The novelty is unclear to me. Maybe the presentation is sub-optimal, but I do not see any novel methodology here or insight. To make this more clear: it is well known that randomized search heuristics work very well on a wide variety of optimization problems (both, combinatorial and numerical). The real open question in this field is to *prove* under which conditions these methods will work, and under which conditions these methods will not work. What is the expected number of objective function evaluations? What is the probability that a solutions that is epsilon-close to a "good" solution is found in polynomial time? For me, answering any of these questions would make the paper at hand acceptable.

3. The formal presentation regarding classic an recent results can be improved (see below).

##########################################################################

Some statements are unfortunate: e.g., on page 2, the authors state "the research community has started questioning the commonly assumed hypothesis if gradient based optimizers get stuck in local minima.". However, this is far from being a "commonly assumed hypothesis". It is a matter of fact from numerical optimization that gradient based optimizers *will*for*sure* get stuck in local minima. The "assumption" is, that these local minima are bad solutions. The authors must be careful with such statements, since many researchers spent decades to reveal insights into numerical optimization which shall not be ignored by today's scientists.

This impreciseness in statements appears for recent works as well: On page 3, the authors state that "weight agnostic neural networks (WANN) also searches for architectures, but keeps the set of weights fixed.". This is, however, not correct. WANNs evaluate the expected performance of a model over various parameters which are shared by all connections. Computing the expectation over multiple parameters is far from keeping weights "fixed".

#########################################################################

---

> ### Author Response · Authors · 2020-11-18
> **Reply to AnonReviewer2**
>
> Thanks for the feedback. While the mentioned citations are related to this work, we are not sure if they are "highly relevant" to this work. We will add the citations which you have mentioned.
>
> We also respectfully disagree that the real open question is to prove under which conditions these methods will work or not work. Whether someone wants to work on proving under which conditions these methods will work or not is a personal preference. There are several papers which make interesting observations about deep neural networks and are published each year in this conference.
>
> As far as we know, we are not aware of any random search heuristics based methods which have shown promising results on image recognition benchmarks like CIFAR10 with deep convolutional neural networks (>= 10 layers). This result is a core contribution of our work. If there are other methods, we would be happy to include those methods in our work if you could share them.
>
> We also feel some concerns regarding the related work are overly critical. The reviewer states that "gradient based optimizers will for sure get stuck in local minima". This is in regards to our statement in the related work about whether or not gradient based optimizers using back-propagation for training neural networks get stuck in local minima. In the paper, in the very next line we cite papers titled "Gradient Descent Provably Optimizes Over-Parameterized Neural Networks" and "Gradient Descent Finds Global Minima of Deep Neural Networks". Both these papers show scenarios where the global minima can be attained. The WANN paper uses a fixed set of weight values ([−2, −1, −0.5, +0.5, +1, +2]) when it searches for the connections in the network and the authors use the same terminology. We do not think any of our statements are imprecise, as mentioned in the review.

---

### Official Review · AnonReviewer3 · 2020-11-02
**Interesting work but with evaluation issues**

**Rating:** 6
**Confidence:** 2

**Review:**

The paper proposes an RSO (random search optimization) method for training deep neural networks. This method is gradient-free and based on the Markov Chain Monte Carlo search. In particular, it adds a perturbation to weight in a deep neural network and tests if it reduces the loss on a mini-batch: the weight is updated if this reduces the loss, and retained otherwise.

Merits of the paper:
+ This paper shows that repeating the RSO process a few times for each weight is sufficient to train a deep neural network. As a result, the number of weight updates is an order of magnitude lesser when compared to backpropagation with SGD.
+ It can make aggressive weight updates in each step as there is no concept of learning rate. The weight update step for individual layers is also not coupled with the magnitude of the loss.
+ RSO is evaluated on classification tasks on MNIST and CIFAR-10 datasets where it achieves competitive accuracies.

Issues of the paper:
- One potential issue is that the method is only evaluated on relatively small networks. I wonder how it works for larger networks such as ResNet-50 and ResNet-101.
- The current figures only show the comparison in terms of training iterations. I would like to see the comparison in terms of training time, ie accuracy-time curve.
- It would be interesting to see the performance of RSO on ImageNet and/or COCO.

---------------------------------------------------------------------------------------------------------------------
I have read the response, and the rating is not changed.

---

> ### Author Response · Authors · 2020-11-18
> **Reply to AnonReviewer3**
>
> Thank you for reviewing our paper. We provide a response to the concerns.
>
> RSO is the first effective attempt to train reasonably deep neural networks (>= 10 layers) with a search based technique on image recognition benchmarks like CIFAR10. We also show that the performance improves as we increase the network depth. One of the highlights of RSO is that explicitly searching for the direction for each weight parameter leads to significantly faster convergence if we count the number of weight updates (which can be as small as 5). Backpropagation on the other hand updates all the weights in the same direction, even though some weights may benefit from moving in a different direction. However, because backpropagation shares the weight update step for all parameters and thus is much faster. Therefore, in its current form, training with RSO with larger networks like ResNet-50 or ResNet-101 and larger datasets like ImageNet or COCO is prohibitively expensive, as the cost of each weight update in RSO scales with the number of network parameters.
>
> Note that RSO can be parallelized across nodes and GPUs, so it is possible to train larger networks and on larger datasets, but this would require hundreds of nodes in its current form. All our experiments were performed on a single GPU. There are several avenues to pursue research on accelerating RSO as mentioned in response to AnonReviewer4 in the “Computational requirements and accelerating RSO” comment.
>
> Please check our response to AnonReviewer1’s question on “Why would it be more meaningful to compare in terms of "cycles"?” for comparisons in terms of training time. We have mentioned the training time per epoch for SGD and RSO below the MNIST and CIFAR10 plots in the paper.

---

### Decision · Program_Chairs · 2021-01-07
**Final Decision**

**Decision:**

Reject

**Comment:**

The paper proposes a variant derivative-free optimization algorithm, that belongs to the family of Evolution Strategies (ES) and zero-order optimization algorithms, to train deep neural networks. The proposed Random Search Optimization (RSO) perturbs the weights via additive Gaussian noise and updates the weights only when the perturbations improve the training objective function. Unlike the existing ES and black-box optimization algorithms that perturb all the weights at once, RSO perturbs and updates the weights in a coordinate descent fashion. RSO adds noise to only a subset of the weights sequentially, layer-by-layer, and neuron-by-neuron. The empirical experiments demonstrated RSO can achieve comparable performance when training small convolutional neural networks on MNIST and CIFAR-10.

The paper contains some interesting ideas. However, there are some major concerns in the current submission:

1) Novelty: there is a wealth of literature in optimization neural networks via derivative-free methods. The proposed algorithm belongs to Evolution Strategies and other zero-order methods, (Rechenberg & Eigen, (1973); Schmidhuber et al., (2007); Salimans et al., (2017). Unforunately, among all the rich prior works on related algorithms, only Salimans et al. (2017) is merely mentioned in the related works. Furthermore, the experiments only compared against SGD rather than any other zero-order optimization algorithms.

Many ideas in Algorithm 1 was proposed in the prior ES literature:

- Evaluate the weights using a pair of noise, -\deltaW and +\deltaW in Alg1 Line13-14 is known as antithetic sampling Geweke (1988), also known as mirrored sampling Brockhoff et al. (2010) in the ES literature.

- Update the weights by considering whether the objective function has improved or not was proposed in Wierstra et al. (2014) that is known as fitness shaping.

Given the current submission, it is difficult to discern the contribution of the proposed method when compared to the prior works. In addition, the convergence analysis of the zero-order optimization was studied in Duchi et. al. (2015) that includes the special coordinate descent version closely related to the proposed algorithm.

2) Experiments:

- Although the experiments showcase the performance of sequential RSO, the x-axis in Figure 4 only reported the iterations after updating the entire network. The true computational cost of the proposed RSO is the #forwardpass x #parameters, that is much more costly than the paper currently acknowledges. Also, RSO requires drawing 5000 random samples and perform forward-passes on all 5000 samples for every single weight update. It will be a great addition to include the #multiplications and computation complexity of RSO and the baseline algorithms.

- More importantly, the paper only compared RSO with SGD in all the experiments. It will significantly strengthen the current paper by including some of the existing ES algorithms.


In summary, the basic idea is interesting, but the current paper is not for publication and will need further development and non-trivial modification.

---

> ### Author Response · Authors · 2021-01-21
> **This review process did not follow ICLR guidelines**
>
> The AC has raised concerns in a completely "new review" posted in the decision letter. While these concerns can be addressed, we were not given the opportunity to do so and this is because due process was not followed by the AC. The ICLR review policy https://iclr.cc/Conferences/2021/ACGuide clearly mentions that the AC should act as a "Discussion Moderator" and "Discussion Participant" and if they had concerns with the paper, they should have posted the concerns during the discussion phase.
>
> ICLR is a conference which distinguishes itself from other conferences and mentions in the review guide "The discussion phase at ICLR is different from most conferences in the AI/ML community. During this phase, reviewers, authors and area chairs engage in asynchronous discussion and authors are allowed to revise their submissions to address concerns that arise. It is crucial that you are actively engaged during this phase." The spirit of this policy was not upheld during this review process.